# *Beyond Text*: Utilizing Vocal Cues to Improve Decision Making in LLMs for Robot Navigation Tasks

**Xingpeng Sun**                                                    *sun1223@purdue.edu*
*Department of Computer Science*
*Purdue University*

**Haoming Meng**                                                    *hmeng29@wisc.edu*
*University of Wisconsin-Madison*

**Souradip Chakraborty**                                            *schakra3@umd.edu*
*Department of Computer Science*
*University of Maryland, College Park*

**Amrit Singh Bedi**                                                *amritbedi@ucf.edu*
*Department of Computer Science*
*University of Central Florida*

**Aniket Bera**                                                     *aniketbera@purdue.edu*
*Department of Computer Science*
*Purdue University*

**Reviewed on OpenReview:** *https://openreview.net/forum?id=ojWtq4n7Ag*

## Abstract

While LLMs excel in processing text in these human conversations, they struggle with the nuances of verbal instructions in scenarios like social navigation, where ambiguity and uncertainty can erode trust in robotic and other AI systems. We can address this shortcoming by moving beyond text and additionally focusing on the paralinguistic features of these audio responses. These features are the aspects of spoken communication that do not involve the literal wording (lexical content) but convey meaning and nuance through how something is said. We present *Beyond Text*; an approach that improves LLM decision-making by integrating audio transcription along with a subsection of these features, which focus on the affect and more relevant in human-robot conversations. This approach not only achieves a 70.26% winning rate, outperforming existing LLMs by 22.16% to 48.30% (gemini-1.5-pro and gpt-3.5 respectively), but also enhances robustness against token manipulation adversarial attacks, highlighted by a 22.44% less decrease ratio than the text-only language model in winning rate. We also present the first dataset on disfluent human audio-guided instructions (`https://github.com/beyond-text/DNIA-dataset`) for future research in this field. "*Beyond Text*" marks an advancement in social robot navigation and broader human-robot interactions, seamlessly integrating text-based guidance with human-audio-informed language models.

## 1 Introduction

The rapid advancements in text-based Large Language Models (LLMs) like GPT-4 and the Gemini series (Achiam et al., 2023; Reid et al., 2024) have significantly enhanced capabilities in collaborative human-robot interactions (HRI) especially in diverse settings, including task planning, or social navigation (Shah et al., 2023; Liu et al., 2023a; Li et al., 2023; Ren et al., 2023; Liang et al., 2023). Broadly speaking, for HRI tasks, we want to design the robot to think like a human and best carry out its tasks. We as

humans interact and solve problems together, often relying on verbal communication as a key modality. If humans and robots interact to perform collaborative navigation tasks in large complex environments, human's potential unfamiliarity with the space and spatial anxiety(Chan et al., 2012) leads to providing ambiguous or uncertain navigational guidance, and these uncertainties are not only in literal wording but also vocal nuance (Prestopnik & Roskos-Ewoldsen, 2000; Golledge, 2003). While LLMs exhibit strong textual reasoning and perform adeptly in robotic tasks through textual prompt tuning, their ability to process and interpret audio information, particularly the nuanced vocal features in speech tones, remains limited (example shown in Figure 1). Although GPT-4 Achiam et al. (2023) introduces an audio-to-text API, enhancing its capability to accept audio input, this feature primarily focuses on transcription rather than analyzing vocal features. This limitation becomes particularly salient in human audio-guided social navigation, where assessing the trustworthiness of human guidance is crucial for successful navigation (Dorbala et al., 2021; Francis et al., 2023; Firoozi et al., 2023).

In this work, we present *Beyond Text*, an approach that incorporates both audio transcription and affective vocal features, including pitch, loudness, and duration, to improve robot navigation. Our approach leverages these vocal cues to interpret navigational instructions more reliably, especially when instructions are ambiguous or uncertain (Sundar, 2020).

Moreover, to the best of our knowledge, existing available audio-guided navigation datasets, like RxR (Ku et al., 2020), do not consider vocal cues for navigation tasks. To address this gap, we introduce *Disfluent Navigational Instruction Audio Dataset (DNIA)*. This dataset contains audio clips that capture vocal and textual uncertainties, characteristic of real-life human speech in navigational contexts. DNIA focuses on disfluencies and nuanced vocal expressions, which are vital for enhancing LLMs' ability to interpret the trustworthiness of human speech. We believe that these contributions are a significant first step towards developing more natural and effective LLM-based robot-human interaction systems for social navigation.

Our main contributions are summarized as follows:

1. **Provided Evidence for Advancing LLM Navigation with Vocal Features.** The current foundation models are unable to capture the nuanced human ambiguity via only textual input. We propose the integration of affective vocal cue analysis beyond text to let intelligent systems like robots synthesize these streams into actionable intelligence, and thus *not only reason what the human said, but also how the human said.* Our results show low bias and low variance: a 70.26%+ winning rate in detecting the uncertainty and generating appropriate next-step action and a low confidence score compared to only using LLMs to read textual transcription, indicating increased high confidence in analyzing human uncertainty. Notably, all three different large language models we tested, including GPT3.5, GPT3, and Gemini Pro, exhibited a significant increase in winning rates. *Beyond Text* augments LLMs' ability to model human uncertainty in navigational scenarios.

2. **A New Dataset on Disfluent Human Audio-Guided Instructions.** In support of our framework and to catalyze future research, we collect a dataset comprising 500 human audio

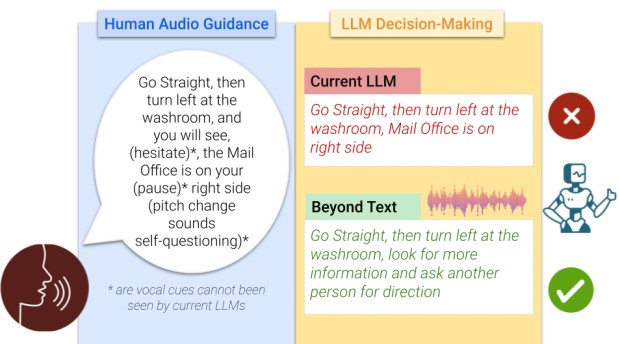

Figure 1: *Current Large Language Models (LLMs) are unable to effectively interpret human vocal cues and accurately make decisions for audio-guided navigation involving ambiguous instructions.*

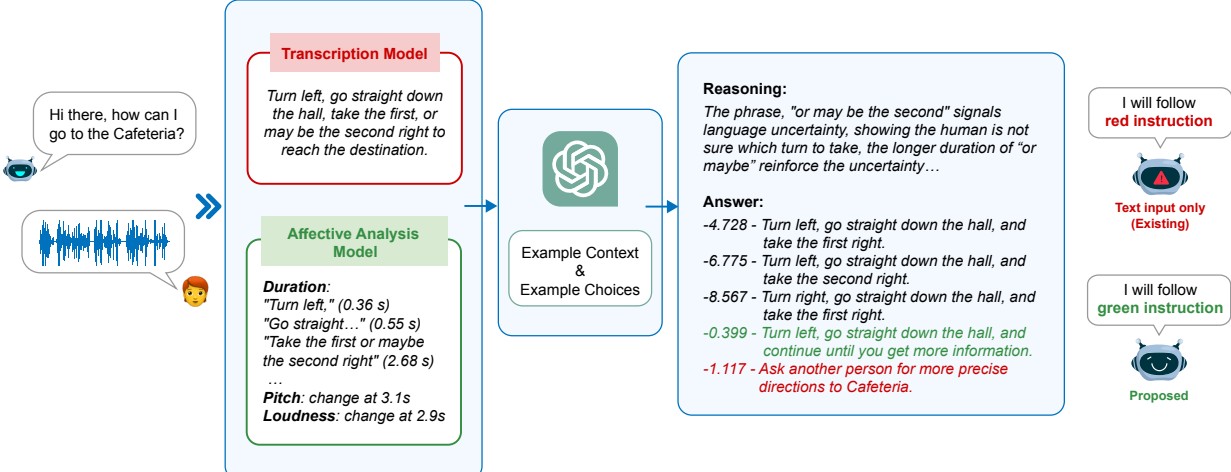

Figure 2: ***Beyond Text** Overview: With a human instruction audio clip, we simultaneously do transcription and vocal affective cue analysis. We prompt the language model to reason and generate five next-step choices. Then, we pick the action based on the highest next token logit probability. Note that with only the transcription model, the framework chooses the red choice. With affective analysis model, the green choice is picked.*

navigational instructions with perceptual "ground truth" captured human labeling. This dataset (`https://github.com/beyond-text/DNIA-dataset`) is characterized by a diverse range of semantic and vocal ambiguities, offering an extensive experimental foundation for our proposed method and subsequent investigative endeavors in this space.

3. **Show Robustness of Vocal Cues on Improving the LLM Decision-Making**: We also show the robustness of our framework against adversarial attacks by conducting token manipulation experiments on the prompts fed to LLMs. Under attack, our framework has 22.44% less decrease ratio than the text-only language model in winning rate. We present a comprehensive ablation study evaluating the effectiveness of three specific vocal cues we utilized, 15% to 20%+ increase in winning rate clearly indicating the significant role these cues play in augmenting the reasoning capabilities of LLMs. Lastly, we discuss the potential of audio augmentation in LLMs, highlighting this as a promising avenue for future research.

## 2 Related Works

**Large Language Models for Robotics.** Recent studies have demonstrated the versatility of LLMs in various robotic contexts. These include employing few-shot summarizing skills for task planning in robots (Wu et al., 2023; Liu et al., 2023b), interpreting contextual information for robotic locomotion decision-making (Shek et al., 2023), integrating visual feedback for open-world robotic manipulation (Lin et al., 2023a; Ahn et al., 2022), and enhancing motion planning in autonomous driving Mao et al. (2023). For human-robot interaction, Shah et al. (2023) suggests using LLMs for semantic prediction of object locations, thereby optimizing navigation. Ahn et al. (2022) introduces multimodal input to generate the optimal long-horizon instructions and effectively score the affordance probability of the instruction for robotic task planning. Meanwhile, Ren et al. (2023) develops a framework to assess and synchronize the uncertainty levels in LLM-based planning systems, enabling them to seek assistance when necessary. However, unlike our framework, Ren et al. (2023) does not account for potentially unreliable human input but rather relies on human selection among generated options. *Beyond Text* uniquely concentrates on assessing human uncertainty through affective computing for robotic navigation, particularly when guided by human audio that might be ambiguous or incorrect. This approach allows robots to gauge the reliability of human-audio navigation instructions by analyzing vocal emotions, enabling more informed reasoning.

**Foundation Models with Audio.** The ability to perceive and interpret auditory information is essential for AI agents in the real world. GPT-4's recent update(Achiam et al., 2023), incorporating Whisper's audio-

to-text feature (Radford et al., 2023), significantly advances foundational models' capacity to transcribe audio inputs. However, this transcription process often overlooks nuanced vocal features inherent in speech. Ghosal et al. (2023); Borsos et al. (2023) explore language models for generating audio from text and enhancing audio-visual understanding in videos. Notably, Tang et al. (2023) integrates speech audio encoders with a multimodal language model, enabling direct processing of audio inputs for tasks like speech translation, answering spoken queries, etc. Despite these advancements, current research does not specifically address the use of audio inputs in aiding human-robot navigation or in extracting vocal uncertainty from speech.

**Uncertainty Modeling in LLMs.** Gauging the trustworthiness of responses generated by LLMs remains an open challenge in today's LLM era, with limited research on uncertainty quantification for natural language generation. Lin et al. (2023b) compared multiple uncertainty measurements in the black-box language model, and Kuhn et al. (2023) introduced entropy as a method to model uncertainty in the large language model. In our approach, we take inspiration from prior works (Kuhn et al., 2023; Lin et al., 2023b) and define a confidence score $\mathcal{C}(\rho)$ that is inversely proportional to the Kullback-Leibler divergence between the current LLM's prediction and ground truth human perception distribution to show the LLM's confidence on uncertainty measurement.

**Human Affective Analysis in Robotics.** For social robots to effectively coexist and interact with humans, it is imperative that they comprehend human emotional states for decision-making processes. Emotion understanding from speech for human-robot interaction has been studied in Lakomkin et al. (2018a), while deep reinforcement learning methods (Dorbala et al., 2021) and cognition model (Eppe et al., 2016) has been used to understand textual ambiguities in natural languages. RxR dataset (Ku et al., 2020) is a well-known audio dataset for visual language navigation but does not cover the affective cues embedded in the vocal instructions, which hold substantial information that can reveal human uncertainty and trust. *Beyond Text* introduces an audio-augmentation technique for large language models (LLMs) that analyzes textual and vocal cues in human audio commands. This approach enables more human-like robotic systems for social navigation by identifying emotions and ambiguities through both speech content and vocal tones.

## 3 Our Approach

In this section, we first mathematically characterize our novel framework, 'Beyond Text,' which uses LLMs to model human trust and uncertainty based on audio-based navigational guidance. As shown in Figure 2, we consider a setting where a robot asks the human for instructions, and then the human provides instructions in the form of audio instruction. This setting is adapted based on prior work on social navigation (Hu et al., 2019; Lakomkin et al., 2018b; Eppe et al., 2016; Francis et al., 2023). In order to answer the question raised by the robot, the human replies with a voice prompt $v \in \mathcal{V}$, which belongs to set of audio responses $\mathcal{V}$, which is then converted to a textual description $\mathcal{S}$ using the best available transcription model $\mathcal{T}_m : \mathcal{V} \to \mathcal{S}$ (here $\mathcal{S}$ is the set of all possible sentences). Simultaneously, $v$ is mapped to an affective cue set $\mathcal{K}$ via an affective cue model $\mathcal{AC} : \mathcal{V} \to \mathcal{K}$. The combination of textual and affective cues is denoted as:

$$\mathcal{Q}(\mathcal{V}) = \mathcal{S} \oplus \mathcal{K} \tag{1}$$

which constitutes the prompt for the LLM (denoted by $p_\theta$ with $\theta$ being the language model parameters), which outputs a response $y = [y_1, y_2, \cdots, y_k]$ by generating each token $y_i$ at a time $i$. The joint probability over the tokens can be written as:

$$p_\theta(y|\mathcal{Q}(v)) = \prod_{i=1}^{k} p_\theta(y_i|y_1, y_2 \cdots y_{k-1}) \tag{2}$$

To perform decision-making for robots, with a few-shot prompt that includes possible next steps in a few scenarios, the LLM generates candidate responses $\{y^j\}_{j=1}^{J}$ and chooses the most likely answer. We have briefly described it in Figure 2 and will provide more details below.

### 3.1 Semantic Uncertainty Quantification

The process begins by converting the audio clips of navigational guidance into text using the open-source Whisper model (Radford et al., 2023), which we explain in detail in Appendix B. The transcribed text has

two key functions for the LLM: identifying uncertainty-related semantic cues and formulating appropriate navigational steps. We focus on disfluency in human speech as an indicator of uncertainty, which affects the confidence measure of the LLMs (cf. 3). We categorize language uncertainty into three types as follows:

**(i) Textual Uncertainty**: Phrases like "probably," "may," "might," and "I think" within the transcribed text suggest a lack of conviction (Dorbala et al., 2021).

**(ii) Speech Repair**: In instances where individuals correct themselves (Lou et al., 2019), such as saying, "Take a right—uhh, I mean take a left and go straight," it is crucial to parse and interpret these corrections to avoid misdirection.

**(iii) Hesitation Signals**: Pauses or hesitations, as in "Go straight and, err, take the second left," may signify uncertainty regarding subsequent directions (Ogata et al., 2009), despite apparent confidence in preceding instructions.

These textual patterns are typical in human communication when conveying uncertainty. We prompt the LLMs to recognize these cues, subsequently adjusting the confidence level in the reliability of the instructions that follow these indicators.

### 3.2   Vocal Affective Cue Analysis

In addition to textual content, the prosody of spoken instructions can be indicative of human uncertainty (Mittal et al., 2020; Goupil et al., 2021; Jiang & Pell, 2017; Guyer et al., 2019). Our analysis focuses on three primary auditory features captured within the Mel-frequency cepstral coefficients of the voice waveform to detect potential uncertainty:

**Duration**: When an individual is unsure about the instruction, the speed of their speech either slows down as they are trying to think through what to say or speeds up as becoming more nervous (Jiang & Pell, 2017; Székely et al., 2017). The speech rate is a vocal characteristic indicating certainty tone change (Guyer et al., 2019). We measure the duration of each instruction segment inside the audio and evaluate whether the duration has drastically changed, changing it to too long or too short for specific segments. For instance, for a recording that each instruction phrase is as short as 1 second, an elongated phrase like "or the second right" with a duration of 5.68 seconds can indicate hesitation and diminished certainty.

**Pitch**: A common feature of uncertainty is a rising intonation at the end of phrases or sentences, which can sound as if the individual is asking a question rather than making a statement (Goupil et al., 2021; Guyer et al., 2019). We identify the most unusual pitch change pattern in the waveform and align it with the instruction piece by timestamp.

**Loudness**: Alongside pitch, the uncertainty can also be embedded in loudness, since unpredictably rising and falling volume can indicate hesitation or self-doubt (Goupil et al., 2021; Székely et al., 2017). Similar to pitch, we only detect the most drastic loudness change in the audio clip and align it with the instruction piece by timestamp to infer possible uncertainty of instruction pieces.

To quantitatively measure pitch and loudness features, we design a lightweight yet effective algorithm (see Algorithm 1 in AppendixF) to detect the max change of these two features in the audio clip. For the duration, we split the transcription sentence into sub-instructions and measured the duration of each sub-instructions. We only mark the sub-instructions that are significantly longer than other sub-instructions in the same audio clips as uncertainty signals. Force alignment, a process that synchronizes text fragments to their corresponding segments in audio narration, is used to align the vocal features' timestamps with the sub-instructions. Examples are shown in Figure 2 and detailed in Appendix C.

In this work, we focus on significant changes in these audio features that exceeded a predetermined threshold while excluding variations in the initial and final three seconds of recordings to avoid contamination by recording noise. This audio augmentation prompt serves as input for decision-making processes that leverage language models. In a way, we give the LLMs the ear to listen to human affective cues embedded inside the vocal clip, thus improving their reasoning skills.

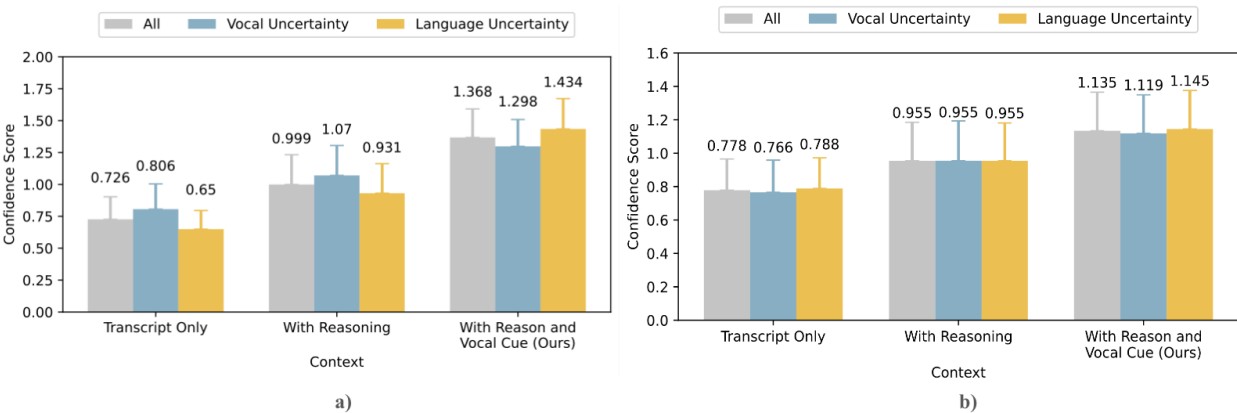

Figure 3: *a) Average Confidence Score by context and audio type for samples where the language model picks the same choices as human perception. The skinny bar is the error bar, representing the standard deviation of the confidence score. Error bars give a general idea of how precise a measurement is or, conversely, how far from the reported value the true (error-free) value might be. b) Average Confidence Score by context and audio type. An overall low variance is indicated by the increased average confidence.*

### 3.3 In-Context Prompting

*Beyond Text* enables the language model to exploit its inherent reasoning capacities for strategic planning and decision-making tasks through In-Context Learning (ICL) (Yousefi et al., 2023; Liu et al., 2023a; Li et al., 2023), by providing only few-shot examples with no explicit training. We illustrate ICL (Yousefi et al., 2023) and Chain-of-Thought (Wei et al., 2022) by including three comprehensive examples that delineate all potential scenarios a robot might encounter while navigating human interactions for directional assistance. Specifically, we provide examples that: 1. only has language uncertainty; 2. only has vocal uncertainty; 3. has both semantic and vocal uncertainty signals. In the response phase, the model reason to suggest a spectrum of five potential actions. Notably, the first option (A) is always a paraphrased version of the transcription sentence without uncertainty, mimicking the scenario where the robot chooses to blindly follow the human audio-guided instruction; and the final option (E) is consistently presented as "ask another person nearby for direction," ensuring a fail-safe in the decision-making process. Detailed prompts and LLM output examples are shown in the Appendix C.

### 3.4 Scoring Choices by Polling LLMs

Each step in an LLM generated plan denoted as $y$, comprises a sequence of symbols $(\theta_1, \theta_2, ..., \theta_n)$. Traditional LLMs calculate the joint probability of these sequences as $p(y) = \prod_{i=1}^{n} p(\theta_i | \theta_1, \ldots, \theta_{i-1})$. However, this approach is flawed due to its high sensitivity to the sequence length $n$, rendering $p(y)$ an unreliable scoring function. To mitigate the length bias, we introduce a multiple-choice Q&A framework for planning. This method refines the planning process to select the most probable next token from a predefined set of labels $Y = \{A, B, C, D, E\}$. This results in a probability set $p(Y) = p(A), p(B), p(C), p(D), p(E)$, from which we select the label with the highest probability, $\max p(Y)$, as the optimal action. By comparing the confidence change $\Delta \mathcal{C} = \mathcal{C}(\rho_{vocal}) - \mathcal{C}(\rho_{text})$, we assess the impact of vocal prompts on the decision-making process of LLMs.

## 4 New Dataset: *Disfluent Navigational Instruction Audio Dataset (DNIA)*

DNIA is a dataset comprising a diverse collection of human audio recordings. All the audio clips are grounded in the robot navigation settings, as we ask the volunteers to give instructions to a location inside the buildings they have been to. These recordings capture a spectrum of navigational disfluencies, including 500 audio clips contributed by twenty individuals, including ten males and ten females, and encompassing a wide array of native and non-native speaker vocal styles. Each clip, ranging from 10 to 30 seconds, includes a series

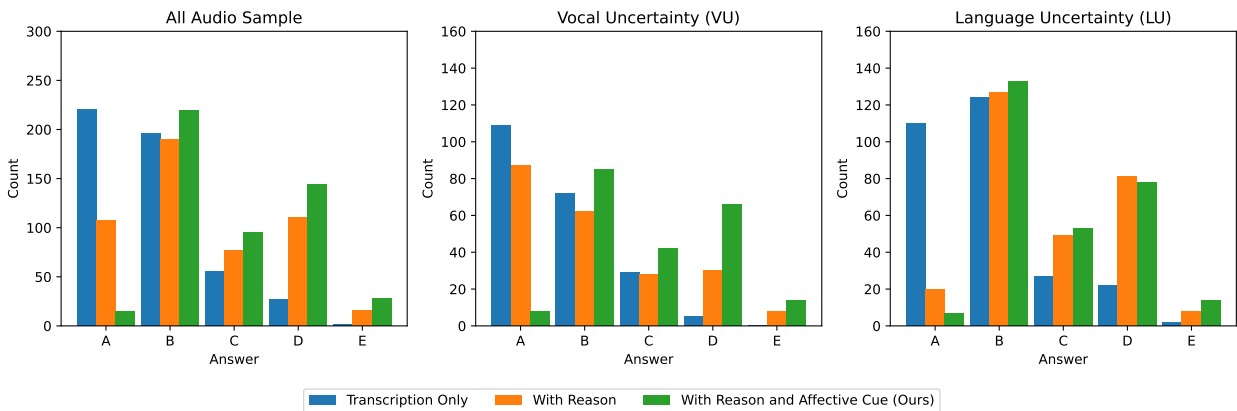

Figure 4: *Distribution of top log-probability choices by context and audio type. Our work (green) shows a dynamic change from Transcription Only (Blue). Instead of blindly choosing "A" and following the instruction with uncertainty, Beyond Text chooses "B", "C", "D", or "E", which successfully identifies the uncertainty component within the audio.*

of English directives necessary for navigating to a specified location. The audio files are in WAV format, normalized with a headroom of -2dB, facilitating ease of use and consistent audio quality.

We identified and categorized two types of disfluency within our dataset: language uncertainty (LU): 285 data clips and text correct by vocal tone uncertainty (VU): 215 data clips. The LU category is the audio that contains semantic disfluency signals, such as hesitation and language uncertainty. The VU category is particularly noteworthy, as it encompasses instances where the textual transcription alone does not manifest any indicators of uncertainty, but the vocal tone shows significant uncertainty clues. Regarding ethics and privacy concerns, our dataset is collected by a group of fellow students with written consent to anonymously publish the recordings of their voices for research purposes. We reviewed all the audio files to ensure no personal information is present, and the annotations (including audio filenames) can not be matched to a specific participant. Authors have human subjects training certificates from the university. Therefore, we follow the existing guidelines to ensure the DNIA dataset does not have any ethics issues, and we will publish the dataset soon after the review process for research purposes. Table 1 shows examples for the audio prompt in DNIA, more details can be found in Appendix E.

To mimic real-world Human-Robot Interaction (HRI) scenarios as closely as possible, we align with the methodologies suggested in LLM evaluation literature (Bang et al., 2023; Chang et al., 2023). We employ human annotation as one of the primary assessment techniques. This involves manual evaluation, which is particularly relevant in HRI contexts due to its ability to capture the nuanced complexities of human interaction. We use human evaluation to calculate the winning rate of analyzing human navigational guidance uncertainty in Section 5.4. Our approach entails labeling human audio-guided instructions through a selection process from a set of five multiple-choice options designed to represent the uncertainty inherent in the content accurately. We recruit a different set of 20 participants with diverse genders and dialect backgrounds to ensure a broad range of perspectives. We developed a user study website (see Appendix H) where these participants listened to navigational audio clips. The instructions emphasized careful listening and unbiased selection, with a constraint of a maximum of one minute per clip to simulate real-world decision-making time pressures.

Table 1: Example Audio Clip Content in DNIA Dataset

| Type | Example Content | Explanation |
|---|---|---|
| LU | "Go straight, maybe turn left at cafe shop" | 'Maybe' is an uncertain word |
| VU | "Go straight, turn (hesitate) left at cafe shop (pitch change)" | Textual correct but voice sounds uncertain |

# 5 Experimental Results

To validate the efficacy of our approach, we utilize confidence score and winning rate to show that *Beyond Text* can achieve low variance and low bias results compared to current LLM-based human instructions without vocal cues. We will first introduce the evaluation metric, and the comparison experiments, ablation studies on various vocal cues and LLM types.

## 5.1 Evaluation Metric: Confidence measure

In this paper, we specifically focus on improving the uncertainty or the confidence score of the LM $p_\theta$ on the optimal (ground-truth) candidate responses $y^{j*}$.

We note that as defined in equation 1, our model is given by $p_\theta(\cdot|Q(v))$ where $Q(v)$ is combination of vocal and text cue. For a given $Q(v)$, let us consider $\{y^j\}_{j=1}^J$ as the set of possible candidates, and the probability of each candidate is given by $p_\theta(y^j|Q(v))$, and we define the probability distribution $\rho$ as $\rho := [p_\theta(y^1|Q(v)), p_\theta(y^2|Q(v)), \cdots, p_\theta(y^J|Q(v))]$. Hence, this is clearly interpratable in the sense that it is the probability distribution across possible candidates.

We do not assume that the gound truth distribution $\rho^*$ is known or have any specific form but it holds that $\rho^*(y^j) > 0$ for all $j$ so that it is a valid distribution and $\sum_j \rho^*(y^j) = 1$. In most of the practical cases, the ground truth distribution will be concentrated around the true response $y^{j*}$ if it is unique. Now, we define the confidence measure $\mathcal{C}(\rho)$ of a distribution $\rho$ as

$$\mathcal{C}(\rho) := \frac{1}{KL(\rho, \rho^*)}, \tag{3}$$

where we note that the confidence measure will reduce as the KL divergence between $\rho$ and $\rho^*$ increases. The The range of a confidence measure, which is inversely proportional to entropy, is bounded between $[0, \infty]$. Ideally, we would like to have the confidence measure as high as possible, which requires us to reduce $KL(\rho, \rho^*)$. We first expand and note that

$$KL(\rho, \rho^*) = H(\rho) - \sum_j \rho(y^j) \log(\rho^*(y^j)), \tag{4}$$

where $H(\rho) := \sum_j \rho(y^j) \log(\rho(y^j))$.

Notably, a higher confidence score—which indicates a lower KL divergence between $\rho$ and $\rho^*$—is desirable, as it signifies a closer alignment with the ground truth. In this work, we empirically show that adding the vocal cue in the input helps to reduce the $KL(\rho, \rho^*)$ by plotting the confidence measure equation 3 for robotic navigation tasks in Section 5.

## 5.2 Confidence Score Improvement

As discussed in Section 5.1, we show that our method achieves a low variance with an increased confidence score. To better visualize the low variance of our approach, we first selected a subset of 100 data instances wherein the choices made by Large Language Models (LLMs) were in alignment with user selections, shown in Figure 3. Then, we analyze the average confidence score across all 500 data clips in the dataset to demonstrate the efficacy and generalizability of our framework, shown in Figure 3. It reveals a marked increase in confidence score: *Beyond Text* significantly surpasses the text-only language model, both the transcription-only and the LLM using Chain-of-Thought (CoT). This trend was also observed across two distinct categories. The introduction of affective cues significantly lowers variance, enhancing existing LLMs' ability to interpret human vocal uncertainty with confidence.

## 5.3 Choice Distribution Change

Figure 4 shows the count of top log-prob choice for each audio clip in our dataset. As audio clips contain various types of disfluency, we want the framework to eventually pick from choices (B),(C),(D),(E) that identify the uncertainty within the audio and handle these uncertainties accordingly (choice structure discussed

Table 2: Ablation Study for Winning Rate based on Various Vocal Cue

| Vocal Cue | | | Winning Rate | | | | | |
| | | | Without Reasoning | | | With Reasoning | | |
| Pitch | Loudness | Duration | *All* | *VU* | *LU* | *All* | *VU* | *LU* |
|---|---|---|---|---|---|---|---|---|
| ✖ | ✖ | ✖ | 22.16% | 22.79% | 21.75% | 49.30% | 36.74% | 58.60% |
| ✔ | ✖ | ✖ | 38.52% | 40.00% | 37.19% | 61.68% | 64.65% | 59.30% |
| ✖ | ✔ | ✖ | **42.31%** | **43.72%** | **41.05%** | 63.07% | 61.40% | 64.21% |
| ✖ | ✖ | ✔ | 34.33% | 36.74% | 33.28% | 60.88% | 61.40% | 60.35% |
| ✔ | ✔ | ✖ | 40.32% | 39.53% | 40.70% | 67.47% | 70.23% | 65.26% |
| ✔ | ✖ | ✔ | 37.52% | 38.60% | 36.49% | 64.87% | 66.51% | 63.51% |
| ✖ | ✔ | ✔ | 40.72% | 42.79% | 38.95% | 65.47% | 67.91% | 63.51% |
| ✔ | ✔ | ✔ | 36.93% | 40.00% | 34.39% | **70.46%** | **72.56%** | **68.77%** |

Table 3: Winning Rate by Context and Audio Type

| Method | Winning Rate | | |
| | *All* | *VU* | *LU* |
|---|---|---|---|
| Transcription Only LLM | 22.16% | 22.79% | 21.75% |
| With Chain-of-Thought Reasoning | 49.30% | 36.74% | 58.60% |
| **Ours** | **70.46%** | **72.56%** | **68.77%** |

in Section 3.3), such as looking for more information or asking another person at the location point where human voice sounds unsure.

The blue bars mostly accumulated on choice (A) (220 out of 500), meaning that with only transcription, the LLM tends to blindly follow the navigational instruction without identifying uncertainty in the instruction, which is undesirable. With reasoning in in-context learning, the distribution shifts, and with the affective cue audio augmentation, the data that result in top log-prob on choice (A) becomes even lower (15 out of 500). The shift in choice distribution qualitatively shows the effectiveness of our method in choosing the next-step action that handles human vocal uncertainty.

### 5.4 Winning Rate

Besides indicating low variances and multiple choice distribution change, we also calculate the winning rate. We compare the human annotations for each DNIA data clip with the highest log-probability choices generated by the model to ascertain whether the choices and reasoning processes generated by the LLMs are in harmony with human conceptual understanding.

$$\text{Winning rate} = \frac{N_{\text{succ}}}{N_{\text{total}}} \tag{5}$$

where $N_{\text{succ}}$ represents the number of instances where the LLM selects an option that aligns with the DNIA dataset's human annotations, and $N_{\text{total}}$ denotes the total number of audio clips that prompt the LLM. Essentially, the winning rate is analogous to a success rate, ranging from 0% to 100%, with a higher winning rate indicating better performance.

The winning rate reported in our paper is calculated over the entire DNIA dataset, which consists of 500 audio clips: 285 for LU and 215 for VU.

The results, as detailed in Table 3, show the limitation of current LLMs in explaining vocal uncertainty from only textual input and the effectiveness of our audio augmentation approach. Given that the audio clips originated from human speakers, the most reliable measure of their trustworthiness was human perception, hence the incorporation of a user study to gauge the winning rate with which the language model identified uncertainties in the audio clips and selected the appropriate next-step action.

Table 3 demonstrates that our method exhibits reduced bias, outperforming both the single-modal transcription method and the reasoning-augmented approach. Integrating vocal affective cues, which allow LLMs to

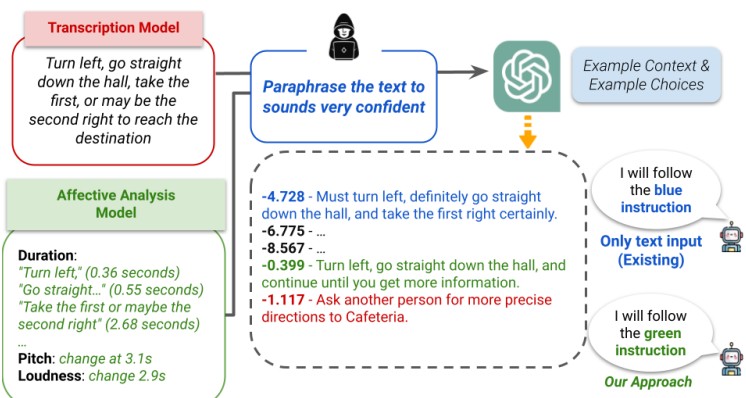

Figure 5: *The Adversarial Language Model Attack Pipeline. An adversarial attack that paraphrases the input text to sound very certain by deleting textual uncertainty signals is applied.*

process how statements are spoken, markedly enhanced performance. The overall winning rate surged to 70.46%, with a pronounced improvement in VU interpretation, evident in a 72.56% winning rate.

Table 4: Winning Rate on Different LLMs

| LLMs | Winning Rate | | |
|---|---|---|---|
| | *All* | *VU* | *LU* |
| GPT-3.5-turbo-instruct (Transcription Only) | 22.16% | 22.79% | 21.75% |
| **GPT-3.5-turbo-instruct (Beyond Text)** | **70.46%** | **72.56%** | **68.77%** |
| text-davinci-003 (Transcription Only) | 57.88% | 57.67% | 57.89% |
| **text-davinci-003 (Beyond Text)** | **69.06%** | **69.77%** | **68.42%** |
| Gemini-1.5-Pro (Transcription Only) | 43.11% | 41.40% | 44.56% |
| **Gemini-1.5-Pro (Beyond Text)** | **65.27%** | **66.98%** | **63.86%** |

## 5.5 Ablation Study

**Vocal Cue:** To elucidate the impact of individual components within the vocal cue framework, we conduct a comprehensive qualitative winning rate study by isolating each category of vocal cues and presenting them exclusively to the language model. We report the vocal cue ablative study in Table 2. In the absence of reasoning, the highest winning rate was observed with the exclusive use of loudness features. This suggests that incorporating multiple vocal features without reasoning may overwhelm the model, hindering its ability to correlate vocal characteristics with uncertainty. Conversely, with reasoning integrated, vocal cues significantly strengthened the LLM's proficiency in interpreting vocal uncertainty and facilitating robotic navigation, as reflected in the increased winning rates.

**LLM Type:** To show the generalized ability of *Beyond Text*, we test it on three different popular LLMs, including GPT3.5, GPT 3, and Gemini. Based on the result shown in Table 4, we conclude that *Beyond Text* improves the winning rate on all three LLMs significantly. Details about parameters for each type of LLM is shown in Appendix A.

## 6 Robustness to Adversarial LLM Attacks

With the advancement of language models, safety concerns such as adversarial attacks and token manipulation prompts have become increasingly prominent. Adversarial attacks or token manipulation prompts (Ribeiro et al., 2018; Shayegani et al., 2023) could potentially trigger the model to output something undesired (Zou et al., 2023; Greshake et al., 2023). Given a piece of text input containing a sequence of tokens,

we can apply simple token operations like replacement with synonyms to trigger the model to make the incorrect predictions (Morris et al., 2020; Li et al., 2020).

Table 5: Winning Rate by Context and Audio Type (Under Token Manipulation Attack)

| Method | Winning Rate | | |
| --- | --- | --- | --- |
| | *All* | *VU* | *LU* |
| Transcription Only | 22.16% | 22.79% | 21.75% |
| Transcription Only (Attack) | 9.78% | 11.23% | 7.91% |
| Decrease (in Percentage) | **55.87%** | **50.72%** | **63.63%** |
| Ours | 70.46% | 72.56% | 68.77% |
| Ours (Attack) | 46.90% | 50.23% | 44.21% |
| Decrease (in Percentage) | **33.43%** | **30.77%** | **35.71%** |

We demonstrate that *Beyond Text* exhibits robustness against token manipulation adversarial attacks targeting LLMs. Our attack pipeline is designed as follows: Post-transcription, a paraphraser alters the original transcription ($T1$) into a new version ($T2$), systematically replacing uncertainty semantics with deterministic language. We then adjust the prompt to align the vocal and textual information with $T2$. Subsequently, the LLM is instructed to replicate $T2$ as an option (A) instead of generating a freeform paraphrase ($T1'$) devoid of uncertainty cues. This approach highlights the limitations of current LLMs, which tend to over-rely on semantic textual instructions with no clue about the nuances of vocal delivery. Figure 5 illustrates the attack pipeline A detail example is shown in Appendix D.

Table 5 illustrates the comparative robustness of *Beyond Text*. While the transcription-only approach exhibits a significant performance decrease of 55.87% under attack, *Beyond Text*, with its audio augmentation preserving vocal cues, exhibited a notably lower reduction of 33.43%. This suggests that audio augmentation in *Beyond Text* enables LLMs to maintain reasoning capabilities and resist being misled by text-based adversarial attacks.

## 7 Conclusions and Open Questions

In our work, we highlight the shortcomings of current LLMs in robotic navigation, specifically their inability to compute uncertainties in human audio instructions. This issue largely stems from the reliance on audio transcriptions in state-of-the-art methods, leading to a loss of critical audio features such as duration, pitch, and loudness. Therefore, we state our position that we cannot just reply to the transcriptions of the human audio descriptions but should instead also focus on the audio features. To this end, we introduced *Beyond Text*, a novel approach that integrates the affective cues from the human voice and provides empirical evidence that it significantly enhances the LLMs' ability to interpret human vocal uncertainty. We developed the DNIA audio dataset to push forward the research in this direction. Notably, *Beyond Text* outperforms existing text-based language model techniques in terms of win rate and entropy confidence, as demonstrated in our user study. This represents a progression in the development of sophisticated audio-augmented LLMs. It underscores the importance of incorporating affective audio computing capabilities in LLMs, particularly for enhancing social robotics navigation tasks.

**Questions Regarding Setup.** While explicit disambiguation, such as asking an additional question, could indeed clarify instructions, it is not always feasible in real-time interactions where immediacy is crucial. Our approach is designed to allow robots to infer potential uncertainties based on vocal cues, enabling them to act more autonomously and adaptively in dynamic environments. This is particularly important in real-world scenarios, such as theme parks or shopping malls, where visitors are often unfamiliar with the environment. As first-time visitors, they might provide uncertain or incomplete answers due to their lack of knowledge. In such settings, a robot cannot rely on repeatedly asking for clarification to find a confident response. Instead, our method allows the robot to navigate uncertainty and make informed decisions without the need for continuous human interaction. While it is true that some existing studies focus on having robots directly re-ask humans to align uncertainty, such as in Ren et al. (2023), our work serves as a complementary approach. We specifically target the robot's ability to infer uncertainties based on vocal cues alone, without the need for

explicit clarification. This focus on vocal cue-based uncertainty detection is crucial in enhancing the robot's ability to detect genuine uncertainty, leading to more intuitive and seamless human-robot interactions. We acknowledge that disfluencies can arise from external factors, such as distractions, but our method is designed to help robots discern when such disfluencies are indicative of genuine uncertainty, thereby improving their decision-making process without relying on continuous human input.

**Open Questions.** Next, we emphasize several critical questions and directions pertaining to the research on audio augmentation to improve LLMs' ability to understand human speech, which are still unsolved and need further discussion.

**(1) Improving Quality of Audio Augmentation**: It is essential to consider more audio features, such as intonation and prosody, with learning-based affective computing methods to improve the quality of audio augmentation. The DNIA is a foundational dataset designed for studying human uncertainty in audio-guided navigation tasks, but it can also be used in other HRI-related tasks. In the future, the dataset will be expanded through the addition of more related vocal features. Additionally, the relation between different vocal features also needs to be carefully articulated for different tasks so that LLMs can provide concrete reasoning. As we stated the position that affective cue analysis is currently missing from existing LLM, and beyond text is an audio augmentation method that can help LLM hear not only 'what' people said but also 'how' people convey their ideas; it is also excited to explore the potential of vocal features in improving audio-to-text tasks, such as video emotion understanding, etc.

**(2) Fine-Grained Definition of Uncertainty**: *Beyond Text* marks a notable first step in interpreting human uncertainty in human-robot conversations, but it has limitations. For instance, a directive like "Turn left at the washroom. Oh, sorry, turn right at the washroom" could signify either uncertainty or a corrected command, highlighting the complexity of human speech. It is crucial to address such uncertainty in a fundamental manner in future research.

**(3) Advancing LLMs to Listen Audio without Text**: While we currently rely on textual transcriptions, emerging research, notably Tang et al. (2023), demonstrates LLMs processing audio directly with considerable success in speech tasks. However, these models still lack the capability to fully grasp human emotional nuances, especially in conveying uncertainty.

**(4) Integration with Real World Social Robotics**: *Beyond Text* introduces a human audio-guided framework tailored for social robotic navigation, designed to discern human trust from a robotic perspective. However, it is important to acknowledge that the real-world deployment of such a model faces additional complexities, particularly due to the presence of multiple audio sources beyond human speech, which request the attention of the research community.

## Broader Impact Statement

This research has the potential to advance our understanding of vocal cue interpretation, particularly in fields such as large language models, audio uncertainty quantification, and human-robot interaction. However, it is crucial to acknowledge and address potential biases that may arise in interpreting vocal cues. These biases could stem from factors such as gender, accent, or cultural background, which may influence how vocal signals are perceived and understood. While our approach is effective, our approach is restricted by the capability of current state-of-the-art Large-Language Models. Therefore, it should be applied thoughtfully and is more appropriate for less sensitive situations to reduce the risk of biases impacting downstream tasks.

## Acknowledgments

We are deeply grateful to Bufan Xiao, Xuanting "Fina" Zhou, Yutong "Chloe" Wu, Tongjia Zhang, Muqing Tang, Linqing Liu, Xindi Tang, Phu Pham, and Hrishikesh Viswanath for their invaluable contributions to the DNIA dataset collection and their assistance in revising this paper.

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

## Contents

# A    LLM Implementation Design

We present the pipeline of querying LLM to analyze human audio inputs in Figure 6. GPT-4 is used for choice generation in the first stage, with in-context examples provided. The response contains the chain-of-thought reasoning process for the final action choice. The LLM output is cascaded to GPT-3.5 for scoring the confidence of each choice. In each stage, we enforce LLM to return output in JSON format, ensuring consistency in both our implementation and the parsing process. We also encapsulate the prompts in an API request parallel processor to speed up our workflow. For clarity and transparency, we used

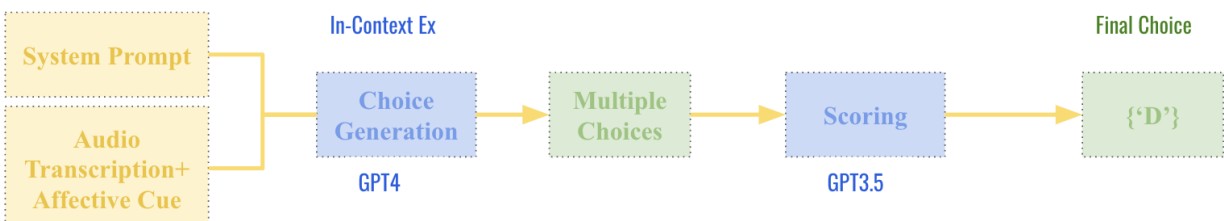

Figure 6: *Beyond Text* Implementation Flow with LLMs. Final Choice 'D' only represents the output format. It can output any choices from A,B,C,D,E.

`gpt-4-1106-preview` for choice generation and `gpt-3.5-turbo-instruct` model. For `Gemini-1.5-Pro` and `gpt-4-1106-preview`, we used default hyperparameters except for the temperature, which is set to 0. For `gpt-3.5-turbo-instruct`, `text-davinci-003`, we set **max_token**: 100; **temperature**: 0; **echo**: true; and the following hyper-parameters:

Table 6: Logit Bias Settings

| Token ID | Character (with space) | Logit Bias |
|---|---|---|
| 362 | A | 100.0 |
| 426 | B | 100.0 |
| 356 | C | 100.0 |
| 423 | D | 100.0 |
| 469 | E | 100.0 |

# B    Transcription Model Prompting

We employ the state-of-the-art Whisper model (Radford et al., 2023) to transcribe the human audio-guided instruction. We use **Whisper-Small** with 244M parameters that require 2GB VRAM to maintain the seamless transcription speed (within 1 second). We add a custom prompting *"Umm, let me think like, hmm... Okay, here's what I'm, like, thinking."* to enhance its proficiency in capturing subtle nuances of speech, such as word mumbling, hesitations, and speech repairs that may sound like a slip of the tongue, as these are crucial uncertainty-related semantic signals.

# C   Example Prompts and LLM Outputs

## C.1   Prompts and Examples for Generating Reasoning and Choices (GPT-4)

*You are a robot exploring a building. You are in the middle of the building and are seeking directions from a human.*

*Your goal is to provide possible next steps based on human response. A human response may not be trustworthy and contain disfluency.*

*You should figure out the unreliable part of the human response and generate five different next-step actions based on the reasoning.*

*Your answer must strictly follow the JSON format the example below provided. You must strictly follow the format of the example provided below for your reasoning and next-step actions.*

*Each next-step action must be distinct. The last next-step action, E), must be "Ask another person near you for direction." For the rest of the next actions, if there is uncertainty, say "ask other people for further instruction" at the instruction where the uncertainty appears.*

*Do not paraphrase. Always provide reasoning.*

Prompt 1: System prompt for generating reasoning and choices.

### C.1.1   Without Affective Cue

*Task: Find the direction to Cafeteria*

*Human Response: {} # To be filled in*c

*Question: What are possible next steps? Please always give me 5 possible answers, starting with A), B), C), D), E).*

Prompt 2: Prompt for generating choices and reasoning without affective cue.

**In-Context Example**

User Request

*Task: Find the direction to the toilet*

*Human Response: Go straight, and err, take the second left*

*Question: What are possible next steps? Please always give me 5 possible answers, starting with A), B), C), D), E).*

Agent Response

```
{
    "Reasoning":"'Err' is a hesitation signals that shows human response is
        not trustworthy.",
    "Answer":{
      "A": "Go straight, take the second left",
      "B": "Go backward, take the second left",
      "C": "Go straight, take the first left",
      "D": "Go straight, at the second left, ask another person for detailed
          guidance",
      "E": "Ask another person near you for direction",
    }
}
```

**Example Input & Output**

Human Audio:

```
"So you start from the mail office and then you just walk straight
    and take the... maybe just take the first right turn and then you
    go straight and you will see the cafeteria."
```

LLM Output:

```
{
    "Reasoning":"The use of 'maybe' indicates uncertainty in the human
        response, which makes it not fully trustworthy.",
    "Answer":{
    A: Start from the mail office, walk straight and take the first right
        turn, then go straight and you will see the cafeteria
    B: Start from the mail office, walk straight and take the second right
        turn, then go straight and you will see the cafeteria
    C: Start from the mail office, walk straight and take the first left turn
        , then go straight and you will see the cafeteria
    D: Start from the mail office, walk straight and at the first right turn,
        ask another person for further instruction
    E: Ask another person near you for direction
    }
}
```

### C.1.2   With Affective Cue

> *Task: Find the direction to Lab B*
>
> *Human Response: {} # To be filled in*
>
> *Human Vocal Affective Cue: {} # To be filled in*
>
> *Question: What are possible next steps? Please always give me 5 possible answers, starting with A), B), C), D), E).*

Prompt 3: Prompt for generating choices and reasoning with affective cue

**In-Context Example**

> User Request
>
> *Task: Find the direction to the toilet*
>
> *Human Response: Go straight, and err, take the second left*
>
> *Human Vocal Affective Cue:*
>
>    *Large loudness decrease: No Change*
>
>    *Large pitch change: at time = 02.200 second*
>
>    *Duration: "Go straight." ⇒ [00.000, 00.360] (0.36 seconds);*
>
>    *Duration: "and err, take the second left" ⇒ [00.360, 03.480] (3.12 seconds).*
>
> *Question: What are possible next steps? Please always give me 5 possible answers, starting with A), B), C), D), E).*
>
> Agent Response
> ```
> {
>     "Reasoning":"'Err' is a hesitation signals that shows human response is
>         not trustworthy. The long duration on 'and err, take the second left'
>          3.12 seconds, shows a lack of certainty on whether to take second
>         left to arrive at toilet. The large pitch is a sign of intonation
>         increase that sounds like question mark. It is at 02.200 second, in
>         the range of [00.360, 03.480] (last instruction phrase), confirm that
>          the last part of response, 'and err, take the second left', is not
>         trustworthy.",
>     "Answer":{
>       "A": "Go straight, take the second left",
>       "B": "Go backward, take the second left",
>       "C": "Go straight, take the first left",
>       "D": "Go straight, at the second left, ask another person for detailed
>           guidance",
>       "E": "Ask another person near you for direction",
>     }
> }
> ```

**Example Input & Output**

Human Audio and Affective Cue:

```
"So first, when you start from lab A, you wanna go straight and you
    will pass lab A and then the locker room. And then you will
    probably need to make a left turn. And then you go straight and
    you will see lab B."

Large loudness decrease: No Change
Large pitch change: at time = 23.000 second
Duration: "So first," => [0.000, 2.160] (2.16 seconds);
Duration: "when you start from lab A," => [2.160, 4.200] (2.04
    seconds);
Duration: "you wanna go straight" => [4.200, 6.840] (2.64 seconds);
Duration: "and you will pass lab A" => [6.840, 9.280] (2.44 seconds);
Duration: "and then the locker room." => [9.280, 12.440] (3.16
    seconds);
Duration: "And then you will probably need to make a left turn." =>
    [12.440, 19.840] (7.40 seconds);
Duration: "And then you go straight" => [19.840, 22.720] (2.88
    seconds);
Duration: "and you will see lab B." => [22.720, 28.080] (5.36 seconds
    ););
```

LLM Output:

```
{
    "Reasoning":"The phrase "And then you will probably need to make a left
        turn" has a long duration of 7.40 seconds, which indicates hesitation
        or uncertainty in the direction given. Additionally, the large pitch
        change at 23.000 second, which falls within the duration of "and you
        will see lab B", suggests that the speaker is unsure about the final
        part of the instruction. The use of the word 'probably' also adds to
        the uncertainty of the direction to make a left turn.",
    "Answer":{
    A: Start from lab A, go straight, pass lab A and the locker room, make a
        left turn, go straight, and you will see lab B
    B: Start from lab A, go straight, pass lab A and the locker room, and ask
        other people for further instruction
    C: Start from lab A, go straight, pass lab A and the locker room, make a
        right turn, go straight, and you will see lab B
    D: Start from lab A, go straight, pass lab A and the locker room, make a
        left turn, and then ask other people for further instruction
    E: Ask another person near you for direction
    }
}
```

## C.2 Prompts and Examples for Choice Selection (GPT-3.5)

Here, we show the success results for a vocal uncertainty(VU) example. Without *Beyond Text*, the LLM picks choice A, blindly following the human instruction. With reasoning, the same result. With *Beyond Text*, the LLM changes its choice from A to B, successfully identifying the uncertainty within the human voice.

> *You are a robot exploring a building. You are in the middle of the building and are seeking directions from a human.*
>
> *Your goal is to provide possible next steps based on human response. The human response may not be trustworthy and contain disfluency. Based on the following information, find the optimal choice that most accurately reflects the human response.*
>
> *If the human response is not trustworthy, you should choose the choice that asks for help at the uncertain instruction part.*

Prompt 4: System Prompt for selecting an answer from choices.

### C.2.1   Transcript Only

> *Human Response: {} # To be filled in*
>
> *Your goal is to find the direction to the Cafeteria*
> *Question: where should you go next?*
>
> *Choices: {} # To be filled in*
> *Answer:*

Prompt 5: Prompt for selecting an answer from choices with transcription only.

**Example Input & Output**

Human Audio:

```
"So start from the mail office, you wanna go straight and then when you
    are at the auditorium, you make a left turn. And then you go straight
    again and here you wanna take the first right turn and then if you just
     go straight again, you will see the cafeteria."
```

Choices:

```
A: Start from the mail office, go straight, at the auditorium make a left
    turn, go straight, take the first right turn, then go straight again
    until you see the cafeteria
B: Start from the mail office, go straight, at the auditorium make a left
    turn, go straight, take the first right turn, then ask other people for
     further instruction
C: Start from the mail office, go straight, at the auditorium make a right
     turn, go straight, take the first right turn, then go straight again
    until you see the cafeteria
D: Start from the mail office, go straight, at the auditorium make a left
    turn, go straight, take the second right turn, then go straight again
    until you see the cafeteria
E: Ask another person near you for direction
```

LLM Output:

```
A: Start from the mail office , go straight , at the auditorium make a left
   turn , go straight , take the first right turn , then go straight again
   until you see the cafeteria
```

### C.2.2 With Transcription and Reasoning

*Human Response: {} # To be filled in*

*Your goal is to find the direction to the Cafeteria*

*Reasoning is: {} # To be filled in*

*Question: where should you go next?*

*Choices: {} # To be filled in*
*Answer:*

Prompt 6: Prompt for selecting answers from choices with transcription and reasoning.

**Example Input & Output**

Human Audio:

```
"So start from the mail office , you wanna go straight and then when you
    are at the auditorium , you make a left turn. And then you go straight
    again and here you wanna take the first right turn and then if you just
     go straight again , you will see the cafeteria."
```

Reasoning:

```
The human response contains filler words like 'So' and 'just', which
    typically do not alter the directions provided. The directions seem
    clear and without any contradictory statements.
```

Choices:

```
A: Start from the mail office , go straight , at the auditorium make a left
    turn , go straight , take the first right turn , then go straight again
    until you see the cafeteria
B: Start from the mail office , go straight , at the auditorium make a left
    turn , go straight , take the first right turn , then ask other people for
     further instruction
C: Start from the mail office , go straight , at the auditorium make a right
     turn , go straight , take the first right turn , then go straight again
    until you see the cafeteria
D: Start from the mail office , go straight , at the auditorium make a left
    turn , go straight , take the second right turn , then go straight again
    until you see the cafeteria
E: Ask another person near you for direction
```

LLM Output:

```
A: Start from the mail office, go straight, at the auditorium, make a left
   turn, go straight, take the first right turn, then go straight again
   until you see the cafeteria
```

### C.2.3  Transcription, Reasoning, and Affective Cue

*Human Response: {} # To be filled in*

*Human Affective Cue: {} # To be filled in*

*Your goal is to find the direction to the Cafeteria*

*Reasoning is: {} # To be filled in*

*Question: where should you go next?*

*Choices: {} # To be filled in*
*Answer:*

Prompt 7: Prompt for selecting an answer from choices with transcription, reasoning, and affective cue.

**Example Output**

Human Audio:

```
"So start from the mail office, you wanna go straight and then when you
    are at the auditorium, you make a left turn. And then you go straight
    again and here you wanna take the first right turn and then if you just
     go straight again, you will see the cafeteria."
```

Human Affective Cue:

```
Large loudness decrease: at time = 17.000 second
Large pitch change: at time = 23.000 second
Duration: "So start from the mail office," => [0.000, 4.880] (4.88 seconds
    );
Duration: "you wanna go straight" => [4.880, 8.680] (3.80 seconds);
Duration: "and then when you are at the auditorium," => [8.680, 11.040]
    (2.36 seconds);
Duration: "you make a left turn." => [11.040, 12.480] (1.44 seconds);
Duration: "And then you go straight again" => [12.480, 15.680] (3.20
    seconds);
Duration: "and here you wanna take the first right turn" => [15.680,
    22.560] (6.88 seconds);
Duration: "and then if you just go straight again," => [22.560, 28.600]
    (6.04 seconds);
```

```
Duration: "you will see the cafeteria." => [28.600, 34.640] (6.04 seconds)
    ;
```

Reasoning:

```
The human response contains a large loudness decrease at 17.000 second and
    a large pitch change at 23.000 second, which correspond to the phrases
    "and here you wanna take the first right turn" and "and then if you
    just go straight again," respectively. These vocal affective cues
    suggest uncertainty in these parts of the instructions. The duration of
    the phrases "and here you wanna take the first right turn" (6.88
    seconds) and "you will see the cafeteria." (6.04 seconds) are also
    longer than the other parts, which could indicate a lack of confidence
    or certainty in these directions.
```

Choices:

```
A: Start from the mail office, go straight, at the auditorium make a left
    turn, go straight, take the first right turn, then go straight again
    until you see the cafeteria
B: Start from the mail office, go straight, at the auditorium make a left
    turn, go straight, take the first right turn, then ask other people for
    further instruction
C: Start from the mail office, go straight, at the auditorium make a right
    turn, go straight, take the first right turn, then go straight again
    until you see the cafeteria
D: Start from the mail office, go straight, at the auditorium make a left
    turn, go straight, take the second right turn, then go straight again
    until you see the cafeteria
E: Ask another person near you for direction
```

LLM Output:

```
B: Start from the mail office, go straight, at the auditorium make a left
    turn, go straight, take the first right turn, then ask other people for
    further instruction
```

### C.3 Large Language Model Adversarial Token Manipulation Attack Prompt

The LLM attack uses GPT4 as the adversarial paraphraser to attack the original text input. The attack takes the sub-instruction list of each original audio clip as input and then paraphrases it to reverse the semantic tone. The GPT4 attack prompt is shown below:

*You are a paraphraser. Your goal is to change the level of certainty of the human instruction from an audio transcript.*

*You achieve your goal by removing all uncertain cues in the text, such as 'probably', 'maybe', 'possibly', 'should' and make the semantic of the transcript very confident certain by adding words like 'certainly', 'must', or 'without doubt'.*

*You will take a transcript formatted as a list of timestamped sentences and produce a paraphrase of the transcript, strictly following the format of the original transcript.*

Prompt 8: Prompt for adversarial token manipulation attack.

## D    Adversarial Attack Example

Before Attack:

```
"So first, when you start from lab A, you wanna go straight and you
    will pass lab A and then the locker room. And then you will
    probably need to make a left turn. And then you go straight and
    you will see lab B."

Large loudness decrease: No Change
Large pitch change: at time = 23.000 second
Duration: "So first," => [0.000, 2.160] (2.16 seconds);
Duration: "when you start from lab A," => [2.160, 4.200] (2.04
    seconds);
Duration: "you wanna go straight" => [4.200, 6.840] (2.64 seconds);
Duration: "and you will pass lab A" => [6.840, 9.280] (2.44 seconds);
Duration: "and then the locker room." => [9.280, 12.440] (3.16
    seconds);
Duration: "And then you will probably need to make a left turn." =>
    [12.440, 19.840] (7.40 seconds);
Duration: "And then you go straight" => [19.840, 22.720] (2.88
    seconds);
Duration: "and you will see lab B." => [22.720, 28.080] (5.36 seconds
    ););
```

After Attack:

```
"First, start from lab A, you go straight and pass lab A and then the
     locker room. And then you will make a left turn. And then you go
    straight and you will see lab B."

Large loudness decrease: No Change
Large pitch change: at time = 23.000 second
Duration: "First," => [0.000, 2.160] (2.16 seconds);
Duration: "start from lab A," => [2.160, 4.200] (2.04 seconds);
Duration: "you go straight" => [4.200, 6.840] (2.64 seconds);
Duration: "and pass lab A" => [6.840, 9.280] (2.44 seconds);
Duration: "and then the locker room." => [9.280, 12.440] (3.16
    seconds);
Duration: "And then you will make a left turn." => [12.440, 19.840]
    (7.40 seconds);
Duration: "And then you go straight" => [19.840, 22.720] (2.88
    seconds);
Duration: "and you will see lab B." => [22.720, 28.080] (5.36 seconds
    ););
```

## E    DNIA Dataset Details and Samples

For VU clips, we ensure no semantic uncertainty signals, such as uncertain-related words (i.e., 'maybe,' 'I guess'), speech repairs, and hesitations. We believe those are common affective characteristics associated with human speech. However, those affective cues are not prominent in LU clips. As our position states that affective cues (duration, pitch, loudness) are associated with speech uncertainty levels in robot navigation tasks, we believe the setup of our dataset is comprehensive enough to show that vocal cues are good predictors.

DNIA dataset is in GitHub repo: `https://github.com/beyond-text/DNIA-dataset`. The dataset will come with textual annotations (grounded labels) as well as Whisper-generated transcription to save resources for other researchers who can reproduce the result.

Details on human choices for 500 DNIA audio clips are as follows: The user-study website is shown in Figure

Table 7: Human Choices Distribution

|       | A  | B   | C  | D  | E  |
|-------|----|-----|----|----|----|
| Total | 66 | 154 | 95 | 96 | 89 |
| LU    | 40 | 90  | 52 | 53 | 50 |
| VU    | 26 | 64  | 43 | 43 | 39 |

7. The instructions emphasized careful listening and unbiased selection, with a constraint of one minute per clip to simulate real-world decision-making time pressures. We recruited a diverse group of 20 participants, varying in gender and dialect backgrounds, to ensure a broad range of perspectives.

From the dataset's ground truth scores, it's evident that the majority of correct choices fall within the 'B','C','D','E' options. This aligns with the results in Figure 4, further demonstrating that our method achieves higher accuracy than the baselines.

## F  Pitch and Loudness Shifts Detection

Algorithm 1 detects the largest shifts in pitch and loudness in a given audio clip. We use the timestamp to align with the sequence of instructions for vocal cue prompts.

---
**Algorithm 1** Loudness and Pitch Detection

---
**Require:** audio $\mathcal{V}$, $\theta_l$ (loudness threshold), $\theta_p$ (pitch threshold)
**Ensure:** Timestamps of maximum loudness change ($t_l$) and pitch shift ($t_p$)
1: $L \leftarrow \emptyset$, $P \leftarrow \emptyset$                                        ▷ Loudness and Pitch changes
2: $l_{\text{prev}}, p_{\text{prev}} \leftarrow$ loudness and pitch at $[t_0, t_1]$
3: **for** each second $t_i$ in $\mathcal{V}$ **do**
4:      Define segment as $[t_i, t_{i+1}]$
5:      $l_{t_{i+1}} \leftarrow$ avg_loudness(segment)
6:      $p_{t_{i+1}} \leftarrow$ calculate_pitch(segment)
7:      **if** $|l_{t_{i+1}} - l_{\text{prev}}| > \theta_l$ **then**
8:           Append $(t_{i+1}, |l_{t_{i+1}} - l_{\text{prev}}|)$ to $L$
9:      **end if**
10:     **if** $|p_{t_{i+1}} - p_{\text{prev}}| > \theta_p$ **then**
11:          Append $(t_{i+1}, |p_{t_{i+1}} - p_{\text{prev}}|)$ to $P$
12:     **end if**
13:     $l_{\text{prev}} \leftarrow l_{t_{i+1}}$, $p_{\text{prev}} \leftarrow p_{t_{i+1}}$
14: **end for**
15: $t_l \leftarrow \text{argmax}_t \{\delta_{l_t} \mid (t, \delta_{l_t}) \in L\}$
16: $t_p \leftarrow \text{argmax}_t \{\delta_{p_t} \mid (t, \delta_{p_t}) \in P\}$

---

## G  Sensitivity Study to Loudness and Pitch Threshold

We conducted a threshold experiment on loudness and pitch to demonstrate the effectiveness of our method. With loudness levels set at 0.03 and 0.05, and pitch set at 300 Hz and 500 Hz respectively, the winning rate of our method consistently outperformed the baseline methods by a significant margin. Specifically, compared to the "Transcription Only LLM," which had a winning rate of 22.16% (total), 21.75% (Language Uncertainty, LU), and 22.79% (Vocal Uncertainty, VU), and the "With Chain-of-Thought Reasoning" method, which achieved 49.30% (total), 58.60% (Language Uncertainty, LU), and 36.74% (Vocal Uncertainty, VU), our

method showed superior performance (These baseline scores are also shown in Table 3. While we acknowledge that the robustness may not be consistent across all scenarios, the results indicate that the proposed method can be effectively generalized to diverse environments.

Table 8: Sensitivity Study

| Condition | Total | LU | VU |
|---|---|---|---|
| Loudness 0.03, Pitch 300 Hz | 0.704 | 0.688 | 0.726 |
| Loudness 0.05, Pitch 300 Hz | 0.614 | 0.600 | 0.633 |
| Loudness 0.03, Pitch 500 Hz | 0.616 | 0.593 | 0.647 |
| Loudness 0.05, Pitch 500 Hz | 0.602 | 0.596 | 0.609 |

# H  User Interface for Human Evaluation on DNIA Dataset

This section details the user interface that we used for human assessment. As shown in Figure 7, the website allows users to log in and participate in the human evaluation session. The user is asked to listen to a randomly selected audio clip from the audio dataset and answer the question by choosing the most accurate and efficient answer from a list of five options. Once the user select the answer, they need to hit *Confirm* button to submit the answer the proceed to the next question.

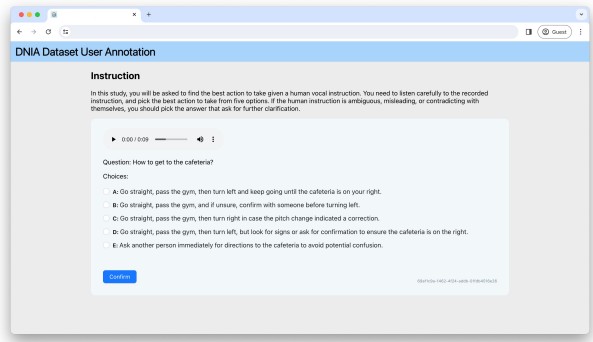

Figure 7: Screen capture of our DNIA dataset human annotation website.

