# OpenReview forum: "Beyond Text: Utilizing Vocal Cues to Improve Decision Making in LLMs for Robot Navigation Tasks"
_TMLR — Accepted by TMLR_

### Review · Reviewer_cPuV · 2024-05-13

**Summary Of Contributions:**

This paper proposes to better use the uncertainty expressed in audio. A given audio is translated into text and an effective analysis model provides additional information about the vocal cues like the duration, the pitch change or the loudness.  This additional information is injected into the prompt. The results show that reasoning and reasoning + the proposed prompt augmentation based on an effective analysis model help choosing the right next-step action. Additionally, the authors say that they plan to release a dataset called Disfluent Human Audio-Guided Instructions which corresponds in 500 human audio navigational instructions with perceptual “ground truth human labeling”. Finally, the authors ran an ablation study to show that their approach is robust to adversarial attacks.

**Audience:**

Yes

**Broader Impact Concerns:**

I do not foresee any ethical implication of the work that would require adding a Broader Impact Statement.

**Claims And Evidence:**

Yes

**Requested Changes:**

**Confidence Measure**

More information on how the confidence measure is computed is critical as there are some inconsistencies in equation (4).

**More information on the dataset**

More information about the dataset is important:
  - The distribution of scores (i.e how many A,B,C,D,E as this as a large impact on the winning rate of the baseline)
  - The inter rater agreement to have a better understanding of the upper bound
  - Full release of the dataset as it is mentioned as a contribution

**False negative rate**

It could be interesting to compute the false negative rate i.e. the amount of time the choice A was not chosen while it was indeed the intended choice. The negative rate could be especially interesting in the case of LU only.

**Minor changes**

Especially in the Related Works, \citep should be used instead of \citet in many places.

**Strengths And Weaknesses:**

### Strengths

- The paper shows strong improvements for the win rate.
- I appreciated the detailed examples in the appendix.

### Weaknesses

**Confidence Measure**

In 5.1, the $\rho$ distribution is not really defined, is it the probability of each possible candidate under $p_\theta$ divided by the sum of $p_\theta$ for all the candidates? This distribution is very sensitive to the possible candidates and not obviously interpretable in between set of candidates.

The quantity in (4) is not a KL divergence. In addition, a true KL between between $\rho$ and $\rho^*$ is not defined as it exists $j$ in $J$ such that $\rho(j) \neq 0$ and $\rho^*(j) = 0$. The KL that exists is KL($\rho^*$,$\rho$).

From the second sentence of 5.2, the confidence score could be KL($\rho^*$,$\rho$). Is that the case? If yes, I do not really understand why it would be called a ‘confidence measure’ or a ‘variance’ as it does not take into account the whole distribution but just the probability weight on the ground truth possibility.

**Improvement on the LU subset with VU information**

In Table 2, on the LU subset, the scores improve with the pitch/loudness/duration information. Would it mean that there is a leak of VU information on the LU subset?

**Dataset**

Lack of some information in the dataset:
  - The distribution of scores (i.e how many A,B,C,D,E as this as a large impact on the winning rate of the baseline)
  - The inter rater agreement to have a better understanding of the upper bound
  - As TMLR does not have a deadline why not waiting to have the dataset released before the submission as the dataset is claimed as a contribution?


**Question about the setup**

Why not simply disambiguate the instruction explicitly instead of implicitly taking into account the uncertainty? Disambiguating the instruction explicitly would simply require asking an additional question to make sure the human is uncertain.

Not clear to me if we want the robots to infer the real instructions and not just follow the instructions given to the robot. The disfluencies can be due to external factors (e.g. distraction from someone) and have nothing to do with the actual uncertainty of the human interacting with the robot.

---

> ### Author Response · Authors · 2024-09-24
> **Revisions Based on Reviewer cPuV Suggestions (1/2)**
>
> Dear reviewer, we appreciate your insightful review and suggestions. The requested changes have been thoroughly addressed and are detailed below.
>
> **Confidence Measure**
>
> 1. We apologize for the confusion and the oversight in not defining $\rho$ explicitly. We have now revised the paper with the definition and are also explaining it here for quick reference. We note that as defined in (2), our model is given by $p_{\theta}(\cdot \mid Q(v))$ where $Q(v)$ is a combination of vocal and text cues. For a given $Q(v)$, let us consider
>
> \begin{align}
> \{y^j\}_{j=1}^{J}
> \end{align}
>
> as the set of possible candidates, and the probability of each candidate is given by $p_{\theta}(y^j \mid Q(v))$. We define the probability distribution $\rho$ as $\rho := [p_{\theta}(y^1 \mid Q(v)), p_{\theta}(y^2 \mid Q(v)), \dots, p_{\theta}(y^J \mid Q(v))]$. Hence, this is clearly interpretable in the sense that it is the probability distribution across possible candidates.
>
>
>
> 2. Thank you for the catch. The current explanation in the paper is not entirely correct. We revise it as follows. We do not assume that the gound truth distribution $\rho^*$ is known or have any specific form but it holds that $\rho^*(y^j)>0$ for all $j$ so that it is a valid distribution and $\sum_j \rho^*(y^j)=1$. In most of the practical cases, the ground truth distribution will be concentrated around the true response $y^{j*}$ if it is unique. Now, we define the confidence measure $\mathcal{C}(\rho)$ of a distribution $\rho$ as
>
> \begin{align}
>     \mathcal{C}(\rho):=\frac{1}{KL(\rho,\rho^*)},
> \end{align}
>
> where we note that the confidence measure will reduce as the KL divergence between $\rho$ and $\rho^*$ increases. The
> the range of a confidence measure, which is inversely proportional to entropy, is bounded between $[0,\infty]$. Ideally, we would like to have the confidence measure as high as possible, which requires us to reduce $KL(\rho,\rho^*)$. We
> first expand and note that
> %
> \begin{align}
>     KL(\rho,\rho^*)=H(\rho)-\sum_{j}\rho(y^j)\log(\rho^*(
> y^j)),
> \end{align}
> where $H(\rho):=\sum_{j}\rho(y^j)\log(\rho(
> y^j))$.
>
>
> 3.  Thank you for your comment. We believe our response to previous comments could clarify these comments as well. We are indeed interested in evaluating $KL(\rho,\rho^*)$ as discussed in detail. Because we don't know our ground truth distribution $\rho^*$, we cannot evaluate terms involving its specific use in practice.
>
>
> **More information on the dataset**
>
> Thank you for the suggestion. The choice distribution of baselines and our method is illustrated in Figure 4 of the paper. Details on human choices for 500 DNIA audio clips are as follows:
> **Total**: A: 66, B: 154, C: 95, D: 96, E: 89
> **LU**: A: 40, B: 90, C: 52, D: 53, E: 50
> **VU**: A: 26, B: 64, C: 43, D: 43, E: 39
>
> We developed a user-study website and instructed raters to follow these guidelines:
>
> "In this study, you will be asked to identify the best action to take given a human vocal instruction. You need to listen carefully to the recorded instruction and select the best action from five options. If the human instruction is ambiguous, misleading, or self-contradictory, you should choose the option that requests further clarification."
>
> The user-study website is shown in Figure 7. The instructions emphasized careful listening and unbiased selection, with a constraint of one minute per clip to simulate real-world decision-making time pressures. We recruited a diverse group of 20 participants, varying in gender and dialect backgrounds, to ensure a broad range of perspectives.
>
> From the dataset's ground truth scores, it's evident that the majority of correct choices fall within the 'B'/'C'/'D'/'E' options. This aligns with the results in Figure 4, further demonstrating that our method achieves higher accuracy than the baselines.
>
> Additionally, the full dataset has been released under an anonymous GitHub repository: [https://anonymous.4open.science/r/DNIA-dataset-D22E](https://anonymous.4open.science/r/DNIA-dataset-D22E), for review purposes only. Once the paper is accepted, we will draft a detailed terms of use statement that requires users to sign before downloading, ensuring that the audio clips are used strictly for research purposes.
>
> **False Negative Rate**
> Following the reviewer's suggestion, we calculated the false negative rate as the percentage of instances where choices A/B/C/D/E were not selected, despite being the intended choices.
>
> - **False negative rate for Total**:
>   A: 3.03%
>   B: 3.25%
>   C: 24.21%
>   D: 18.75%
>   E: 2.25%
>
> - **False negative rate for LU**:
>   A: 2.5%
>   B: 3.33%
>   C: 25%
>   D: 26.42%
>   E: 2%
>
> - **False negative rate for VU**:
>   A: 3.85%
>   B: 3.13%
>   C: 23.26%
>   D: 9.3%
>   E: 2.56%
>
> Overall, these results validate the reliability of our approach in accurately selecting the intended choices.

---

> ### Author Response · Authors · 2024-09-24
> **Revisions Based on Reviewer cPuV Suggestions (2/2)**
>
> **Question About the Setup**
>
> Thank you for raising this point. While explicit disambiguation, such as asking an additional question, could indeed clarify instructions, it is not always feasible in real-time interactions where immediacy is crucial. Our approach is designed to allow robots to infer potential uncertainties based on vocal cues, enabling them to act more autonomously and adaptively in dynamic environments. This is particularly important in real-world scenarios, such as theme parks or shopping malls, where visitors are often unfamiliar with the environment. As first-time visitors, they might provide uncertain or incomplete answers due to their lack of knowledge. In such settings, a robot cannot rely on repeatedly asking for clarification to find a confident response. Instead, our method allows the robot to navigate uncertainty and make informed decisions without the need for continuous human interaction.
> While it is true that some existing studies focus on having robots directly re-ask humans to align uncertainty, such as in the paper "Robots That Ask For Help: Uncertainty Alignment for Large Language Model Planners," our work serves as a complementary approach. We specifically target the robot's ability to infer uncertainties based on vocal cues alone, without the need for explicit clarification. This focus on vocal cue-based uncertainty detection is crucial in enhancing the robot's ability to detect genuine uncertainty, leading to more intuitive and seamless human-robot interactions. We acknowledge that disfluencies can arise from external factors, such as distractions, but our method is designed to help robots discern when such disfluencies are indicative of genuine uncertainty, thereby improving their decision-making process without relying on continuous human input.
>
> **Minor Changes**
>
> Thanks for the suggestion. We will make sure to change the related work’s \citep to \citet in the final version.
>
>
> We once again appreciate your thorough review and will incorporate these updates into the final version of the paper once it is accepted.

---

### Review · Reviewer_FxnN · 2024-09-10

**Summary Of Contributions:**

The authors note that LLMs have shown promise as decision-making tools for robot navigation, but pure text can drop relevant information from human instructions. Assuming a context where people speak to robots, the proposed method is to use audio cues rather than just the transcribed audio. An example of an audio cue is the duration spent on each sentence, to check if the speaker is speaking quickly or slowly.

The focus on the audio cues is to determine locations where the person providing instruction may be uncertain. To use these audio cues, the LLM does not take audio natively - instead it takes a summary of audio cues as additional part of text. This includes large changes in volume, larges changes in pitch, and the time spent saying each sentence of text. Transcription is done using Whisper, slightly prompted to do better at parsing mumbled text.

The resulting method is found to lead to better success rates on two datasets constructed to have vocal uncertainty (VU) and language uncertainty (LU), with largest gains on the VU dataset, indicating vocal cues can be helpful.

**Audience:**

Yes

**Broader Impact Concerns:**

It is a more minor point but I do wonder if there are potential fairness concerns for people who are less able to vocally express themselves, tend to speak with more vocal uncertainty, etc.

**Claims And Evidence:**

Yes

**Requested Changes:**

Would appreciate a clearer definition of winning rate - winning rate relative to what? How many data points is the comparison over?

**Strengths And Weaknesses:**

The paper is generally clear on the implementation, I appreciate having direct transcripts of the prompts and responses because it was not actually that clear from the maintext how this was done. The ablations between chain-of-thought alone vs chain-of-thought + audio cues is appreciated.

The paper not using any direct multimodal LLM that natively takes audio does harm some of the conclusions (although this is mentioned in the conclusions as an open question).

---

> ### Author Response · Authors · 2024-09-24
> **Revisions Based on Reviewer FxnN Suggestions**
>
> Thank you for your considerate review and suggestions. We have addressed the requested changes as detailed below.
>
>
> We apologize for the confusion. The winning rate is defined as (also mentioned in word in Section 5.4 in the original paper):
>
> $ \text{Winning rate} = \frac{N_{\text{succ}}}{N_{\text{total}}}$
>
> where $N_{\text{succ}}$ represents the number of instances where the LLM selects an option that aligns with the DNIA dataset’s human annotations, and $N_{\text{total}}$ denotes the total number of audio clips that prompt the LLM. Essentially, the winning rate is analogous to a success rate, ranging from 0% to 100%, with a higher winning rate indicating better performance.
>
> The winning rate reported in our paper is calculated over the entire DNIA dataset, which consists of 500 audio clips: 285 for LU and 215 for VU.
>
>
> Thank you for this valuable suggestion. We have now included a section addressing broader impact concerns.
> This research has the potential to advance our understanding of vocal cue interpretation, particularly in fields such as large language models, audio uncertainty quantification, and human-robot interaction. However, it is crucial to acknowledge and address potential biases that may arise in interpreting vocal cues. These biases could stem from factors such as gender, accent, or cultural background, which may influence how vocal signals are perceived and understood. While our approach is effective, our approach is restricted by the capability of current state-of-the-art Large-Language Models. Therefore, it should be applied thoughtfully and is more appropriate for less sensitive situations to reduce the risk of biases impacting downstream tasks.
>
> We appreciate your thorough review and will incorporate these updates into the final version of the paper once it is accepted.

---

### Review · Reviewer_odF2 · 2024-09-13

**Summary Of Contributions:**

This research paper investigates the use of vocal cues to improve LLMs for robot navigation tasks. The authors argue that current methods struggle with interpreting ambiguous and uncertain instructions given through audio, often missing the nuances conveyed in a person's voice. To address this, they propose a new approach called "Beyond Text" which integrates audio transcription with an analysis of affective vocal cues like pitch, loudness, and duration. This allows the LLM to better understand the confidence and uncertainty in human instructions, leading to more accurate decision-making for robots. They also introduce a new dataset called Disfluent Navigational Instruction Audio Dataset (DNIA) specifically designed to capture vocal and textual uncertainties present in real-life navigational guidance. Through experiments, they demonstrate that Beyond Text significantly outperforms existing LLM-based approaches for human audio-guided navigation, showing robustness against adversarial attacks and showcasing the potential of audio augmentation for LLMs in human-robot interaction.

**Audience:**

Yes

**Broader Impact Concerns:**

A Broader Impact Statement is needed for this paper. For example, potential biases in interpreting vocal cues (like those stemming from gender, accent, or cultural background) need to be carefully discussed.

**Claims And Evidence:**

Yes

**Requested Changes:**

Required:

1. It is suggested to conduct sensitivity study to the parameters of loudness threshold and pitch threshold in Algorithm 1.

Suggested:

2. Suppose one uses an audio-to-text tool to convert the audio to text first and then uses a text-to-audio tool to convert the text to audio. Then they input the generated audio to the proposed method. It would be interesting to see how the proposed method works in this case. Will the proposed method still show performance gain?

**Strengths And Weaknesses:**

Strengths:

1. The paper effectively argues that existing text-based LLMs fall short in interpreting the nuances of human language, particularly in social navigation tasks where vocal cues play a crucial role in conveying uncertainty.  To address this, the paper introduces Beyond Text, a framework that incorporates both audio transcription and affective vocal features (pitch, loudness, duration) to improve robot navigation, particularly in situations where instructions are ambiguous.

2. To support this approach, the paper introduces the Disfluent Navigational Instruction Audio Dataset (DNIA), addressing the gap in existing datasets by providing audio clips capturing vocal and textual uncertainties characteristic of real-life human speech in navigational contexts.

3. The paper is well-presented and the experiments are comprehensive, demonstrating the effectiveness of the proposed method (I'm not an expert in this specific domain).

Weaknesses:

The main weakness is that the proposed method is mainly based on heuristics and there is no theoretical analysis. Moreover, the paper follows a strong assumption that the duration, pitch, loudness reflect the confidence of the speaker. This assumption might not hold in other cases. Therefore, it might be hard to valid its effectiveness in other datasets or changing environments.

---

> ### Author Response · Authors · 2024-09-24
> **Revisions Based on Reviewer odF2 Suggestions**
>
> Thank you for your thoughtful review and suggestions. We have addressed the requested changes as outlined below.
>
> 1. We conducted a threshold experiment on loudness and pitch to demonstrate the effectiveness of our method. With loudness levels set at 0.03 and 0.05, and pitch set at 300 Hz and 500 Hz respectively, the winning rate of our method consistently outperformed the baseline methods by a significant margin. Specifically, compared to the "Transcription Only LLM," which had a winning rate of 22.16% (total), 21.75% (Language Uncertainty, LU), and 22.79% (Vocal Uncertainty, VU), and the "With Chain-of-Thought Reasoning" method, which achieved 49.30% (total), 58.60%  (Language Uncertainty, LU), and 36.74% (Vocal Uncertainty, VU), our method showed superior performance (These baseline scores are also shown in Table 3 in the paper). While we acknowledge that the robustness may not be consistent across all scenarios, the results indicate that the proposed method can be effectively generalized to diverse environments.
>
> ### Results:
>
> | Condition                      | Total   | LU      | VU      |
> |--------------------------------|---------|---------|---------|
> | Loudness 0.03, Pitch 300 Hz     | 0.704   | 0.688   | 0.726   |
> | Loudness 0.05, Pitch 300 Hz     | 0.614   | 0.600   | 0.633   |
> | Loudness 0.03, Pitch 500 Hz     | 0.616   | 0.593   | 0.647   |
> | Loudness 0.05, Pitch 500 Hz     | 0.602   | 0.596   | 0.609   |
>
>
> 2. Thank you for your suggestion. It's an intriguing idea, but it lies outside the primary focus of our paper. The effectiveness of the proposed method in this scenario would heavily depend on the capabilities of the audio-to-text and text-to-audio tools used. Our method is specifically designed to analyze vocal cues embedded within human instructions to enhance the decision-making process of large language models (LLMs). Current audio-to-text and text-to-audio tools may degrade or entirely lose these vocal cues, leading to audio clips that lack the necessary vocal context. Meanwhile, our paper showcases that the proposed method can effectively capture the language uncertainty cues in the semantic instructions, and still show performance gains in such cases by a large margin that baseline methods. So, under the case you suggested, the problem changed into more language uncertainty only, and our proposed method still shows performance gain.
>
> 3. Thank you for this valuable suggestion. We have now included a section addressing broader impact concerns.
> This research has the potential to advance our understanding of vocal cue interpretation, particularly in fields such as large language models, audio uncertainty quantification, and human-robot interaction. However, it is crucial to acknowledge and address potential biases that may arise in interpreting vocal cues. These biases could stem from factors such as gender, accent, or cultural background, which may influence how vocal signals are perceived and understood. While our approach is effective, our approach is restricted by the capability of current state-of-the-art Large-Language Models. Therefore, it should be applied thoughtfully and is more appropriate for less sensitive situations to reduce the risk of biases impacting downstream tasks.
>
> We appreciate your thorough review and will incorporate these updates into the final version of the paper once it is accepted.

---

### Decision · Action_Editor_nfdN · 2024-10-16

**Recommendation:** Accept as is

**Comment:**

The reviewers were all satisfied with the experiments and ablations conducted in the paper to support the need for vocal cues as part of the LLM for ambiguous navigation tasks. The reviewers also appreciated the release of the supporting dataset.

Most of the reviewer-author discussion had to do with a few additional experiments and some clarifications (e.g., metric definitions). In the end, all reviewers were happy with the paper for acceptance to TMLR.

**Audience:**

Yes, demonstrating the benefit of incorporating vocal cues into navigation tasks would be interesting to the TMLR community. Moreover, the paper presents an open source dataset with vocal information for navigation settings.

**Claims And Evidence:**

The paper proposes using vocal cues to improve LLMs for robot navigation tasks. To address the limitations of using text-based LLMs in navigation, where vocal cues play an important role in conveying uncertainty, the paper introduces Beyond Text, a framework that incorporates both audio and vocal features (pitch, loudness, duration) to improve robot navigation in ambiguous situations. Additionally, the authors present the Disfluent Navigational Instruction Audio Dataset (DNIA), which, unlike other datasets, contains audio clips capturing vocal and textual uncertainties representative of human speech in navigational contexts.

Across the board, reviewers found the claims made in the paper to be soundly supported by evidence.